# 🎬 CML-BENCH: A FRAMEWORK FOR EVALUATING AND ENHANCING LLM-POWERED MOVIE SCRIPTS GENERATION

## ABSTRACT

Large Language Models (LLMs) have demonstrated remarkable proficiency in generating highly structured texts. However, while exhibiting a high degree of structural organization, movie scripts demand an additional layer of nuanced story-telling and emotional depth—the 'soul' of compelling cinema—that LLMs often fail to capture. To investigate this deficiency, we first curated **CML-Dataset**, a dataset comprising *(summary, content)* pairs for Cinematic Markup Language (CML), where 'content' consists of segments from esteemed, high-quality movie scripts and 'summary' is a concise description of the content. Through an in-depth analysis of the intrinsic multi-shot continuity and narrative structures within these authentic scripts, we identified three pivotal dimensions for quality assessment: *Dialogue Coherence (DC)*, *Character Consistency (CC)*, and *Plot Reasonable-ness (PR)*. Informed by these findings, we propose the **CML-Bench**, featuring quantitative metrics across these dimensions. CML-Bench effectively assigns high scores to well-crafted, human-written scripts while concurrently pinpointing the weaknesses in screenplays generated by LLMs. To further validate our benchmark, we introduce **CML-Instruction**, a prompting strategy with detailed instructions on character dialogue and event logic, to guide LLMs to generate more structured and cinematically sound scripts. Extensive experiments validate the effectiveness of our benchmark and demonstrate that LLMs guided by CML-Instruction generate higher-quality screenplays, with results aligned with human preferences. Our work offers a comprehensive framework for both evaluating and guiding LLMs in screenplay authoring.

## 1 INTRODUCTION

Large Language Models (LLMs) have shown strong performance in various creative text generation tasks, from poetry to narrative fiction Radford et al. (2019); Brown et al. (2020); Raffel et al. (2020); Touvron et al. (2023); Grattafiori et al. (2024); Bai et al. (2023a); Yang et al. (2024); Team (2023); Team et al. (2024a;b). However, movie screenwriting presents unique challenges. Unlike general text, movie scripts have a strict structure defined by specific formatting rules (scene headings, action lines, dialogue blocks) and require careful organization of story elements, character development, and plot progression. While current LLMs can generate scripts that follow basic structural rules Gervás (2013); Isaza & Kopp (2025); Mirowski et al. (2023); Kumaran et al. (2023); He et al. (2023); Zheng et al. (2024), they often fail to create scripts with the essential qualities that make movies engaging - emotional depth, thematic meaning, and narrative coherence Liu et al. (2024); Tian et al. (2024); Rashkin et al. (2020); Hueth (2019); Mahon & Lapata (2024); Wang et al. (2025). This quality gap drives our research into new methods for evaluating and improving LLM-generated screenplays.

To systematically investigate the quality gap and understand what constitutes a well-written screenplay, we first constructed a dataset of human-authored scripts, termed the **CML-Dataset**. Starting from the training set of MovieSum Saxena & Keller (2024), which contains approximately 1,800 movie scripts, we filtered this collection using IMDb ratings Maas et al. (2011) and selected 100 classic films with high scores and broad genre coverage All scripts were standardized to the MovieSum format. Given the length and complexity of the original scripts, we employed the Qwen model Yang et al. (2024)

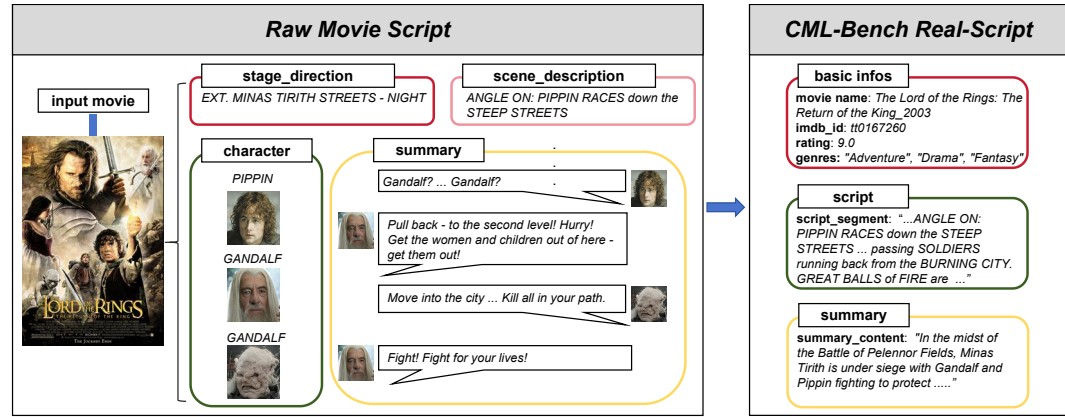

Figure 1: **CML-Dataset construction pipeline**: Raw movie script elements (e.g., stage direction, scene description, character, dialogue) are extracted and transformed into a structured CML-Dataset entry containing basic information, the script segment (content), and an LLM-generated summary.

to automatically extract coherent multi-shot narrative units of 15 to 20 scenes from each script with token distribution statisticized in Figure 7. For each selected segment (the "content"), we used a large language model with carefully designed prompts (see Appendix) to generate a concise summary that captures the main events, characters, and actions, resulting in 100 *(Movie Name, IMDB ID, Content, Summary)* pairs with 600K tokens in total. These high-quality, human-written scripts serve as the ground truth for subsequent analysis and evaluation of LLM-generated scripts with summaries in Sec 5. To further identify the intrinsic characteristics that define high-quality screenwriting, we systematically analyzed the CML-Dataset along three key dimensions: *Dialogue Coherence (DC)*, *Character Consistency (CC)*, and *Plot Reasonableness (PR)*. For DC, we examined the thematic continuity of adjacent dialogue turns; for CC, we assessed the stability and authenticity of each character's linguistic style and emotional expression; and for PR, we evaluated the logical and causal progression of key actions and events. As shown in Sec 3, through linguistic analyses and comparing scripts from highly rated and poorly rated movies, we empirically validated that DC, CC, and PR are pivotal for script quality.

Building on these empirical insights, we introduce **CML-Bench** (Cinematic Markup Language Benchmark), a comprehensive evaluation framework designed to quantitatively assess screenplay quality along the three core dimensions. CML-Bench comprises eight interpretable metrics: DC1 measures the semantic similarity of adjacent dialogue turns, DC2 quantifies topic concentration within dialogues, and DC3 evaluates the relevance of question-answer pairs; CC1 assesses the emotional stability of each character, CC2 measures the consistency of their linguistic style, and CC3 examines the alignment between a character's stated intentions and subsequent actions; PR1 captures the semantic coherence of event sequences, while PR2 quantifies the density of causal relationships between key plot events. Each metric is implemented through a combination of structured parsing, language model-based feature extraction, and embedding similarity calculation, enabling fine-grained and objective evaluation more than simply LLM-based scoring. As illustrated in Figure **??**, CML-Bench reveals that, under base prompting, all seven LLMs consistently underperform compared to human-written scripts across all core dimensions, with particularly large gaps in dialogue coherence, character consistency, and plot reasonableness. This phenomenon highlights the persistent deficiencies of current LLMs in generating high-quality, cinematically sound screenplays.

To address the deficiencies highlighted by CML-Bench and to further validate our benchmark, we introduce **CML-Instruction**. CML-Instruction is a prompting strategy that provides LLMs with detailed instructions on character dialogue and event logic, guiding them to generate more structured and cinematically sound scripts. Extensive experiments validate the effectiveness of this approach. As shown in Figure 4 and Table 1, LLMs guided by CML-Instruction achieve substantial improvements across all CML-Bench metrics. To validate the robustness of CML-Bench across diverse cinematic styles, we conduct case studies (see Figure 5) on scripts from various genres within the CML-Dataset. Furthermore, ablation studies (Figure **??**) are performed to assess the impact of specific components within CML-Instruction on the script generation capabilities of seven different LLMs. Finally,

user studies (Table 2) confirm a strong correlation between CML-Bench evaluations and human preferences.

Our contributions can be summarized as follows: (1) We propose **CML-Dataset**, a specialized dataset of 100 *(summary, content)* pairs derived from classic movie scripts, serving as a high-quality ground truth for screenplay analysis and evaluation. (2) Through analysis of the CML-Dataset, we empirically establish *Dialogue Coherence (DC)*, *Character Consistency (CC)*, and *Plot Reasonableness (PR)* as pivotal dimensions for assessing screenplay quality. (3) We introduce **CML-Bench**, a comprehensive evaluation framework comprising eight interpretable quantitative metrics designed to assess these three core dimensions, enabling fine-grained analysis of LLM-generated scripts. (4) We develop **CML-Instruction**, a prompting strategy that provides LLMs with detailed, component-level instructions, significantly improving their ability to generate structured and cinematically sound screenplays, as validated by CML-Bench and human evaluations.

## 2 RELATED WORKS

**LLM-powered Cinematic Contents Creation**    Large Language Models have demonstrated remarkable capabilities across natural language tasks Radford et al. (2019); Brown et al. (2020); Raffel et al. (2020); Touvron et al. (2023); Grattafiori et al. (2024); Bai et al. (2023a); Yang et al. (2024); Team (2023) and are increasingly applied to creative content generation, particularly visual storytelling He et al. (2023); Lin et al. (2023); Long et al. (2024); Xie et al. (2024); Zheng et al. (2024); Zhao et al. (2024); Blattmann et al. (2023); Khachatryan et al. (2023). Movie scriptwriting has advanced through hierarchical planning (Re$^3$ Yang et al. (2022b), DOC Yang et al. (2022a)), adapted for screenplays in Dramatron Mirowski et al. (2023), multi-agent dialogue in IBSEN Han et al. (2024), and causal plot refinement in R2 Lin et al. (2025). However, effective guidance requires understanding intrinsic screenplay characteristics Rashkin et al. (2020); Hueth (2019); Wang et al. (2024); Liu et al. (2024). Screenwriting literature emphasizes narrative structure, character arcs, dialogue function, and thematic development Field (2005); McKee (1997); Stuart (1999), highlighting that effective screenplays are carefully constructed narratives designed for a visual medium. By identifying key dimensions from actual movie scripts, CML-Bench provides a structured framework for evaluating LLM-generated content and instruction-level guidance to help LLMs produce more cinematically sound and narratively compelling screenplays.

**Evaluation Methods for Movie Scripts**    Traditional NLG metrics like BLEU Papineni et al. (2002), ROUGE Ganesan (2018), and perplexity struggle with creative long-form content, prioritizing lexical overlap over semantic and structural screenplay quality Novikov et al. (2018); Clark & Smith (2021). Long-form content evaluation Kryscynski et al. (2021); Kryscynski (2021); Louis & Nenkova (2013); Durmus et al. (2020); Chen et al. (2024); Yuan et al. (2024) often lacks domain-specific nuances for screenplays. Existing benchmarks like LongBench Bai et al. (2023b) and targeted probes Liu et al. (2023a); Mohtashami & Jaggi (2023) reveal LLMs' challenges with extended context, highlighting the need for specialized creative evaluation frameworks. While "LLM-as-a-judge" shows strong correlation with human judgments Gu et al. (2024); Li et al. (2024); Farchi et al. (2024); Liu et al. (2023b); Chiang et al. (2024); Myung et al. (2024); Zhong et al. (2024), studies indicate LLMs often lack domain-specific understanding for creative content Paech (2023); Zhong et al. (2022); Shen et al. (2025); Saha et al. (2025), potentially overlooking expert-valued qualities. Our CML-Bench addresses these limitations by providing interpretable metrics derived from established screenwriting principles Field (2005); McKee (1997); Stuart (1999). It leverages LLMs for sophisticated feature extraction (e.g., question-answer relevance, causality) but grounds the final assessment in these structured, rule-based evaluations to ensure objectivity and provide actionable feedback on screenplay quality.

## 3 MOVIE SCRIPTS ANALYSIS AND DATASET CONSTRUCTION

### 3.1 DATASET CONSTRUCTION

A movie script is a structured document that serves as the blueprint for film production. It consists of several essential components:

- **Scene Headings (Sluglines):** Indicate the location (INT./EXT.), setting, and time of day (DAY/NIGHT), providing the spatial and temporal context for each scene.
- **Action Lines:** Describe the visual actions, settings, and character movements, conveying what the audience sees and hears, excluding dialogue.
- **Character Cues:** Specify which character is speaking, typically centered on the page.
- **Dialogue:** The spoken words of the characters, indented beneath the corresponding cue.
- **Parentheticals:** Brief instructions or descriptions related to dialogue delivery or action, placed in parentheses below the character cue.
- **Transitions:** Instructions for editing between scenes (e.g., FADE IN, CUT TO, DISSOLVE TO), guiding the flow of the narrative.

To construct a high-quality evaluation benchmark, we curated a dataset from the training set of MovieSum Saxena & Keller (2024), which originally contains approximately 1,800 movie scripts. We filtered this pool using IMDb scores Maas et al. (2011), selecting 100 classic films with high ratings and broad genre coverage. For each selected movie, we extracted the script and standardized its format to match the MovieSum convention. The data preparation process is illustrated in Figure 1. Given that original scripts are often lengthy and contain multiple narrative arcs, we employed the Qwen large language model to automatically identify and extract a segment of 15 to 20 scenes, denoted as $\mathcal{C} = \{c_1, c_2, \ldots, c_N\}$, where $N = 100$. Each $c_i$ is a multi-shot narrative unit that can be summarized into a complete story. This process ensures that each excerpt is both coherent and representative of the original work. The distribution

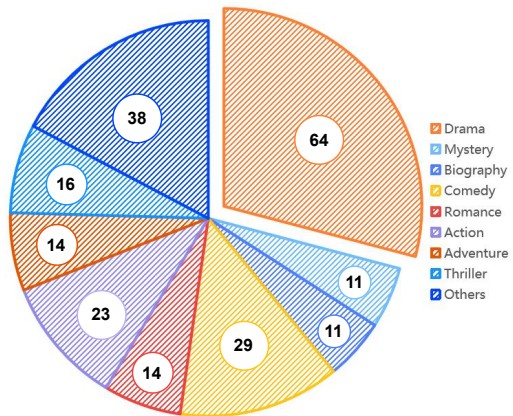

Figure 2: **Genre distribution in CML-Dataset.** The dataset includes all major movie types.

of token lengths for both script segments and summaries is presented in Figure 7, demonstrating that our dataset maintains diversity in content while controlling for length.

Formally, our dataset consists of $N = 100$ tuples $(m_i, s_i, c_i, a_i)$, where $m_i$ is the movie name, $s_i$ is the IMDb ID, $c_i$ is the selected script content, and $a_i$ is the corresponding summary. These 100 human-written script contents serve as the ground truth for all subsequent evaluation and analysis, and the genre distribution is shown in Figure 2

To assess the ability of large language models to generate high-quality screenplays, we use the summary $a_i$ of each movie as input and employ seven different LLMs (see Table 1) to generate scripts. The details of the model selection and generation process are provided in Section 5. The generated scripts are then compared against the ground truth contents $c_i$ using our proposed evaluation metrics.

This curated dataset provides a reliable foundation for analyzing script structure and evaluating the performance of large language models in screenplay generation.

## 3.2 INTRINSIC CHARACTERISTICS ANALYSIS

To empirically identify the core quality dimensions that distinguish high-quality screenplays, we conducted a systematic analysis of the curated human-written script dataset. Our goal is to move beyond superficial structural features and quantitatively capture the intrinsic properties that underpin effective cinematic storytelling. We focus on three fundamental dimensions: dialogue, character, and logic in plots. For each dimension, we designed specific analytical procedures:

1. **Dialogue Coherence (DC):** We posit that the topic of a well-written script dialogue should be highly consistent. Therefore, it is necessary to conduct a thematic consistency analysis of adjacent dialogues.
2. **Character Consistency (CC):** We argue that a character in a script is only considered three-dimensional and vivid if the language they use corresponds to the emotional traits they

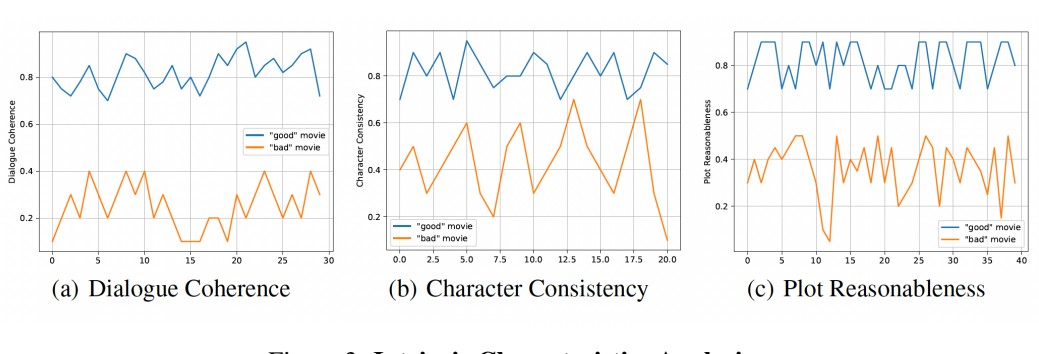

(a) Dialogue Coherence      (b) Character Consistency      (c) Plot Reasonableness

Figure 3: **Intrinsic Characteristics Analysis**.

are meant to convey. Thus, a linguistic and emotional consistency analysis of a character's speech is required.

3. **Plot Reasonableness (PR)**: We believe that a character's actions should be consistent with the current scene to logically advance the plot. Therefore, it is essential to analyze whether a character's behavior is logically consistent with the ongoing events.

To validate these criteria, a comparative analysis was performed on script segments from two distinct movie groups based on IMDb ratings: one cohort with ratings above 9 (indicative of high quality) and another with ratings below 3 (indicative of low quality), applying the aforementioned analytical methods and visualizing the results with the Gemini 2.5 Pro model. As depicted in Figure **??**, scripts from the high-quality group consistently exhibited strong dialogue, character, and plot, whereas scripts from the low-quality group showed marked deficiencies in these areas; this empirical investigation therefore confirms that Dialogue Coherence (DC), Character Consistency (CC), and Plot Reasonableness (PR) serve as pivotal dimensions for assessing screenplay quality, directly informing the design of our evaluation metrics and the construction of the CML-Bench benchmark.

## 4 CINEMATIC MARKUP LANGUAGE BENCHMARKING (CML-BENCH)

### 4.1 AUTOMATED QUANTITATIVE EVALUATION METRICS

To overcome the deficiencies of generic text evaluations and the limitations of current LLM-as-a-judge approaches in capturing domain-specific nuances for screenplay assessment, the CML-Bench framework offers an objective and fine-grained quantitative evaluation. This framework is structured around nine interpretable metrics, systematically organized into three principal dimensions—*Dialogue Coherence (DC)*, *Character Consistency (CC)*, and *Plot Reasonableness (PR)*—which are implemented using a combination of structured parsing, LLM-based information extraction (Gemma Team et al. (2024b) for lightweight feature extraction and Qwen Yang et al. (2024) for complex analysis), and embedding-based similarity calculation. Consequently, CML-Bench provides a reproducible and detailed analysis designed to capture the essential qualities of compelling cinematic writing, moving beyond superficial assessments.

#### 4.1.1 DIALOGUE COHERENCE (DC)

The first dimension, Dialogue Coherence (DC), is designed to capture the logical and topical flow of conversations within a script. Let $\mathcal{D} = [d_1, d_2, \ldots, d_N]$ denote the ordered sequence of dialogue turns in a script segment.

**DC1: Adjacent Turn Topic Similarity.** For each pair of adjacent dialogue turns $(d_i, d_{i+1})$, we compute their semantic embeddings $\mathbf{e}_i, \mathbf{e}_{i+1}$ using the Gemma model. The metric is defined as the mean cosine similarity:

$$\text{DC1}(\mathcal{D}) = \frac{1}{N-1} \sum_{i=1}^{N-1} \frac{\mathbf{e}_i \cdot \mathbf{e}_{i+1}}{\|\mathbf{e}_i\| \, \|\mathbf{e}_{i+1}\|}. \tag{1}$$

A higher value indicates greater topical continuity between turns.

**DC2: Dialogue Topic Concentration.** For each $d_i$, Gemma is used to extract a set of keywords $K_i$. Let $P(w)$ be the empirical distribution of all keywords $w$ across the segment. The topic concentration is measured by the normalized entropy:

$$\text{DC2}(\mathcal{D}) = 1 - \frac{H(P)}{\log |\mathcal{V}|}, \quad H(P) = -\sum_{w \in \mathcal{V}} P(w) \log P(w), \tag{2}$$

where $\mathcal{V}$ is the vocabulary of extracted keywords. Lower entropy (higher DC2) reflects more focused dialogue.

**DC3: Linguistic Creativity.** Creative language features (metaphors, unique expressions, innovative word usage) are extracted from dialogue using Gemma. For each feature $f_i$, Qwen generates a creativity analysis $a_i$, and embeddings $\mathbf{e}_{a_i}$ are computed. The linguistic creativity is measured by inverting the mean pairwise cosine similarity between these analysis embeddings:

$$\text{DC3}(\mathcal{D}) = 1 - \frac{2}{L(L-1)} \sum_{i=1}^{L-1} \sum_{j=i+1}^{L} \frac{\mathbf{e}_{a_i} \cdot \mathbf{e}_{a_j}}{\|\mathbf{e}_{a_i}\| \, \|\mathbf{e}_{a_j}\|}, \tag{3}$$

where $L$ is the number of extracted creative features. Lower similarity (higher creativity) yields higher scores.

### 4.1.2 CHARACTER CONSISTENCY (CC)

The second dimension, Character Consistency (CC), evaluates the stability and distinctiveness of character behavior and language.

**CC1: Character Emotional Stability.** For each character $c$, let $E_c = [e_1, \ldots, e_{N_c}]$ be the sequence of classified emotions (e.g., mapped to $\{-1, 0, 1\}$ for negative, neutral, positive) for their dialogue turns. The emotional stability is measured as:

$$\text{CC1}(c) = 1 - \frac{1}{N_c - 1} \sum_{i=1}^{N_c - 1} |e_{i+1} - e_i|/2. \tag{4}$$

The overall metric is the mean over all characters. Higher values indicate smoother, more plausible emotional arcs.

**CC2: Character Originality.** For each character $c$, distinctive speech features are extracted using Gemma, then analyzed by Qwen to produce uniqueness embeddings $\mathbf{u}_c$. The originality is measured by combining inter-character dissimilarity and intra-character consistency:

$$\text{CC2} = \lambda_1 \cdot \left( 1 - \frac{2}{K(K-1)} \sum_{i=1}^{K-1} \sum_{j=i+1}^{K} \frac{\mathbf{u}_i \cdot \mathbf{u}_j}{\|\mathbf{u}_i\| \, \|\mathbf{u}_j\|} \right) + \lambda_2 \cdot \frac{1}{K} \sum_{c=1}^{K} S_c, \tag{5}$$

where $K$ is the number of characters, $S_c$ is the intra-character dialogue similarity for character $c$, and $\lambda_1 + \lambda_2 = 1$. Higher values indicate characters are both distinctive from each other and internally consistent.

**CC3: Action-Intention Alignment.** Intention-expressing dialogues are identified for each character, and their embeddings are compared to those of subsequent action descriptions. The metric is the mean of the maximal cosine similarity between each intention and any action embedding:

$$\text{CC3} = \frac{1}{L} \sum_{k=1}^{L} \max_a \frac{\mathbf{u}_k \cdot \mathbf{w}_a}{\|\mathbf{u}_k\| \, \|\mathbf{w}_a\|}, \tag{6}$$

where $\mathbf{u}_k$ is the embedding of the $k$-th intention, $\mathbf{w}_a$ is an action embedding, and $L$ is the number of intentions.

### 4.1.3 PLOT REASONABLENESS (PR)

The third dimension, Plot Reasonableness (PR), quantifies the logical structure and plausibility of the narrative.

**PR1: Event Sequence Semantic Coherence.** Key events or scene descriptions $[s_1, \ldots, s_K]$ are extracted (using Qwen or tags), and their embeddings are computed. The metric is the mean cosine similarity between adjacent events:

$$\text{PR1} = \frac{1}{K-1} \sum_{i=1}^{K-1} \frac{\mathbf{s}_i \cdot \mathbf{s}_{i+1}}{\|\mathbf{s}_i\| \, \|\mathbf{s}_{i+1}\|}. \tag{7}$$

Higher values indicate more coherent event progression.

**PR2: Event Coherence.** Qwen is used to extract key plot events from scenes and actions, producing a chronological event sequence. The coherence is measured by the mean cosine similarity between adjacent event embeddings, reflecting logical event progression.

**PR3: Narrative Innovation.** Narrative structure patterns (plot devices, storytelling techniques, structural innovations) are extracted from scenes and actions using Gemma. For each pattern $p_i$, Qwen analyzes its innovation potential to produce embeddings $\mathbf{n}_i$. The narrative innovation is measured by inverting the weighted combination of pattern similarity metrics:

$$\text{PR3} = 1 - \left[ \lambda_3 \cdot \frac{2}{P(P-1)} \sum_{i=1}^{P-1} \sum_{j=i+1}^{P} \frac{\mathbf{n}_i \cdot \mathbf{n}_j}{\|\mathbf{n}_i\| \, \|\mathbf{n}_j\|} + \lambda_4 \cdot \frac{1}{P} \sum_{i=1}^{P} \frac{\mathbf{n}_i \cdot \bar{\mathbf{n}}}{\|\mathbf{n}_i\| \, \|\bar{\mathbf{n}}\|} \right], \tag{8}$$

where $P$ is the number of narrative patterns, $\bar{\mathbf{n}}$ is the centroid of pattern embeddings, and $\lambda_3 + \lambda_4 = 1$. Lower similarity (higher innovation) yields higher scores.

### 4.2 CML-Instruction: Structured Prompting for Script Generation

To address the structural and narrative deficiencies identified by CML-Bench, we propose **CML-Instruction**, a prompting strategy that guides large language models to generate structured movie scripts using detailed, component-level instructions. As a complex instruction, CML-Instruction provides explicit guidance, specifying requirements for scene organization, character dialogue, action descriptions, and other screenplay elements. In particular, the instructions emphasize the consistency and depth of character dialogue, the logical flow of events, and the use of cinematic conventions such as scene headings and transitions. By incorporating these fine-grained instructions, CML-Instruction enables LLMs to better capture the hierarchical and semantic structure of professional screenplays as demonstrated in our experiments in Section 5 and more details are provided in Appendix E.

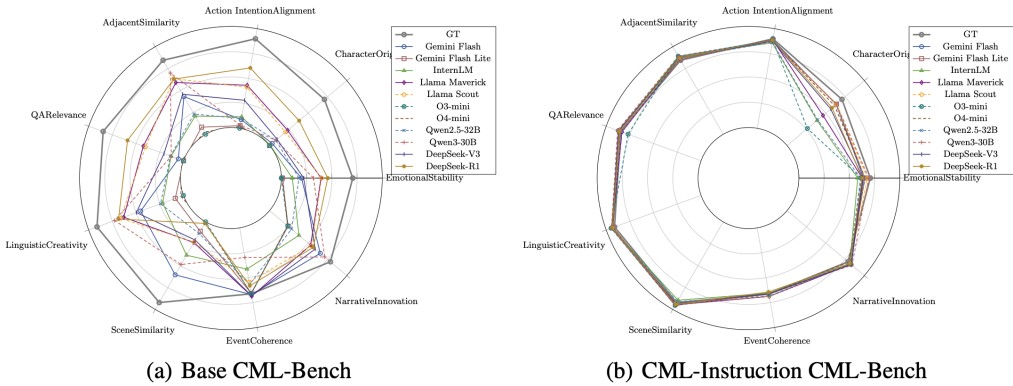

(a) Base CML-Bench      (b) CML-Instruction CML-Bench

Figure 4: **CML-Bench evaluation of LLM-generated screenplays.** (a) Human (GT) vs. LLMs with base prompt settings. (b) Human (GT) vs. LLMs with CML-Instruction.

## 5 Experiments

### 5.1 Main Results

With experimental settings provided in Appendix F, the main experimental results are presented in Table 1. We report the performance of eleven leading LLMs under base and CML-Instruction

Table 1: Main results on the CML-bench. Each model is evaluated on Dialogue Coherence (DC), Character Consistency (CC), and Plot Reasonableness (PR), with sub-metrics DC1–3, CC1–3, PR1–3. The top block shows base LLM generations; the bottom (shaded) block shows results with CML-instruction prompting.

| Model | Dialogue Coherence | | | Character Consistency | | | Plot Reasonableness | | |
|---|---|---|---|---|---|---|---|---|---|
| | DC1 | DC2 | DC3 | CC1 | CC2 | CC3 | PR1 | PR2 | PR3 |
| Human (GT) | 0.85 | 0.85 | 0.91 | **0.71** | 0.284 | **0.90** | 0.92 | 0.90 | 0.39 |
| o3-mini-base | 0.00 | 0.00 | 0.00 | 0.00 | 0.000 | 0.00 | 0.00 | 0.90 | 0.12 |
| o4-mini-base | 0.03 | 0.03 | 0.03 | 0.01 | 0.003 | 0.02 | 0.01 | 0.90 | 0.12 |
| llama-4-maverick-base | 0.59 | 0.42 | 0.63 | 0.39 | 0.094 | 0.43 | 0.24 | **0.91** | 0.27 |
| llama-4-scout-base | 0.63 | 0.40 | 0.66 | 0.40 | 0.082 | 0.41 | 0.23 | 0.86 | 0.27 |
| gemini-flash-base | 0.43 | 0.05 | 0.45 | 0.20 | 0.013 | 0.09 | 0.60 | 0.90 | 0.33 |
| gemini-flash-lite-base | 0.08 | 0.00 | 0.09 | 0.01 | 0.000 | 0.03 | 0.11 | 0.90 | 0.12 |
| internlm3-8b-base | 0.21 | 0.13 | 0.22 | 0.11 | 0.003 | 0.12 | 0.38 | 0.82 | 0.19 |
| qwen2.5-32b-base | 0.23 | 0.14 | 0.24 | 0.18 | 0.032 | 0.10 | 0.13 | 0.87 | 0.14 |
| qwen3-30b-base | 0.70 | 0.13 | 0.73 | 0.32 | 0.033 | 0.03 | 0.49 | 0.79 | 0.36 |
| deepseek-v3-base | 0.46 | 0.21 | 0.50 | 0.22 | 0.039 | 0.28 | 0.21 | 0.90 | 0.30 |
| deepseek-r1-base | 0.63 | 0.59 | 0.69 | 0.46 | 0.153 | 0.61 | 0.02 | 0.87 | 0.28 |
| o3-mini-instr | **0.89** | 0.77 | **0.95** | 0.62 | 0.105 | 0.89 | **0.95** | 0.90 | 0.41 |
| o4-mini-instr | 0.87 | 0.86 | 0.93 | 0.66 | 0.246 | 0.88 | **0.95** | 0.90 | 0.40 |
| llama-4-maverick-instr | 0.87 | 0.83 | 0.94 | 0.63 | 0.186 | 0.89 | **0.95** | **0.91** | 0.41 |
| llama-4-scout-instr | 0.87 | 0.86 | 0.94 | 0.68 | 0.254 | 0.88 | 0.93 | **0.91** | 0.41 |
| gemini-flash-instr | 0.88 | **0.87** | 0.94 | 0.64 | 0.254 | 0.89 | 0.94 | 0.90 | 0.41 |
| gemini-flash-lite-instr | 0.86 | 0.85 | 0.93 | 0.65 | 0.256 | 0.88 | 0.94 | 0.90 | 0.41 |
| internlm3-8b-instr | **0.89** | **0.87** | 0.94 | 0.58 | 0.155 | 0.86 | 0.89 | 0.90 | 0.41 |
| qwen2.5-32b-instr | 0.87 | 0.84 | 0.94 | 0.61 | 0.158 | 0.86 | 0.91 | 0.91 | 0.40 |
| qwen3-30b-instr | **0.89** | 0.88 | **0.95** | 0.69 | **0.255** | 0.88 | **0.95** | 0.90 | **0.420** |
| deepseek-v3-instr | 0.87 | 0.87 | 0.94 | 0.63 | 0.226 | 0.88 | 0.93 | 0.90 | **0.420** |
| deepseek-r1-instr | 0.87 | 0.86 | 0.94 | 0.63 | 0.233 | 0.89 | 0.94 | 0.89 | 0.41 |

(see Sec 4.2) prompting settings, alongside human-written scripts in **CML-Dataset** described in Sec 3.1. Each model is evaluated on the three core dimensions of script quality, with enhanced metrics including **CC2: Character Originality** (measuring character distinctiveness and consistency) and **PR3: Narrative Innovation** (measuring creative storytelling patterns), expanding our evaluation from 8 to 9 comprehensive metrics.

Figure 4 shows human scripts achieve high CML-Bench scores while base LLMs perform poorly. Table 1 reveals: (1) Base models struggle with creativity metrics (near-zero CC2 scores, PR3 varies 0.117-0.357); (2) Many models score zero due to structural failures; (3) Qwen3-30B achieves highest PR3 score (0.420), exceeding human ground truth (0.394). CML-Instruction significantly improves all metrics, with instruction-tuned models approaching human-level coherence but showing room for improvement in character originality.

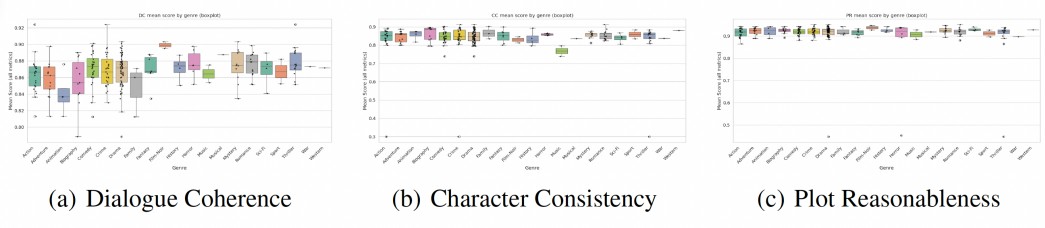

|  (a) Dialogue Coherence  |  (b) Character Consistency  |  (c) Plot Reasonableness |

Figure 5: **Case studies on diverse movie genres** (a) xxx. (b) xxx. (c) xxx.

Table 2: User Study Results: Correlation between CML-Bench scores and human ratings for scripts generated by different models. Human raters evaluated a random subset of scripts on Dialogue Coherence (DC), Character Consistency (CC), and Plot Reasonableness (PR).

| Model | CML-Bench Score | | | Human Rating (Avg.) | | |
|---|---|---|---|---|---|---|
| | DC | CC | PR | DC | CC | PR |
| Human (GT) | 0.87 | 0.84 | 0.91 | 3.00 | 2.66 | 3.33 |
| o3-mini-base | 0.00 | 0.00 | 0.45 | 1.67 | 1.21 | 1.21 |
| o4-mini-base | 0.03 | 0.02 | 0.46 | 1.46 | 1.33 | 1.21 |
| llama-4-maverick-base | 0.55 | 0.43 | 0.58 | 1.42 | 1.29 | 1.25 |
| llama-4-scout-base | 0.56 | 0.43 | 0.55 | 1.13 | 1.04 | 1.42 |
| gemini-flash-base | 0.31 | 0.14 | 0.75 | 1.46 | 1.33 | 1.29 |
| gemini-flash-lite-base | 0.06 | 0.01 | 0.51 | 1.29 | 1.21 | 1.25 |
| internlm3-8b-base | 0.19 | 0.12 | 0.60 | 1.25 | 1.67 | 1.21 |
| qwen2.5-32b-base | 0.23 | 0.18 | 0.50 | 1.49 | 1.57 | 1.46 |
| qwen3-30b-base | 0.70 | 0.32 | 0.64 | 1.51 | 1.82 | 1.55 |
| deepseek-v3-base | 0.46 | 0.22 | 0.56 | 1.78 | 1.63 | 1.36 |
| deepseek-r1-base | 0.63 | 0.46 | 0.45 | 1.66 | 1.79 | 1.48 |
| o3-mini-instr | 0.86 | 0.76 | 0.93 | 3.50 | 3.33 | 3.25 |
| o4-mini-instr | 0.89 | 0.82 | 0.93 | 3.58 | 3.58 | 3.33 |
| llama-4-maverick-instr | 0.88 | 0.81 | 0.93 | 3.50 | 3.42 | 3.50 |
| llama-4-scout-instr | 0.89 | 0.83 | 0.92 | 3.42 | 3.42 | 3.50 |
| gemini-flash-instr | 0.90 | 0.82 | 0.92 | 3.50 | 3.42 | 3.58 |
| gemini-flash-lite-instr | 0.88 | 0.82 | 0.92 | 3.42 | 3.50 | 3.50 |
| internlm3-8b-instr | 0.90 | 0.77 | 0.90 | 3.17 | 3.25 | 3.42 |
| qwen2.5-32b-instr | 0.87 | 0.61 | 0.91 | 3.42 | 3.39 | 3.44 |
| qwen3-30b-instr | 0.89 | 0.69 | 0.93 | 3.49 | 3.35 | 3.51 |
| deepseek-v3-instr | 0.87 | 0.63 | 0.92 | 3.51 | 3.48 | 3.42 |
| deepseek-r1-instr | 0.87 | 0.63 | 0.92 | 3.57 | 3.55 | 3.39 |

## 5.2 COMPREHENSIVE ANALYSIS

**Case Study**   To validate the robustness of CML-Bench across diverse cinematic styles, we present case studies on human-written scripts from various genres in the CML-Dataset (see Figure 2 for genre distribution). Figure 5 illustrates the boxplot distributions for Dialogue Coherence (DC), Character Consistency (CC), and Plot Reasonableness (PR) scores across these genres. These results demonstrate that CML-Bench consistently assigns high scores to quality scripts across the genre spectrum, affirming its robust applicability for evaluating diverse screenplay types.

**Human Evaluation**   To validate CML-Bench's reliability, human experts rated script excerpts from five randomly sampled movies on **DC**, **CC**, and **PR** using a 0-5 integer scale, involving ten participants. Our analysis in Table 2 reveals a strong alignment between CML-Bench scores and human judgments, evidenced by an overall Spearman correlation of 0.80 between averaged CML-Bench scores and averaged human ratings (details in Appendix G.2). This significant correlation confirms that CML-Bench's automated metrics effectively capture human perceptions of screenplay quality, establishing it as a robust and interpretable benchmark.

## 6 CONCLUSION

We introduce a comprehensive framework for evaluating and generating high-quality movie screenplays using large language models. Our contributions include the CML-Dataset (100 classic script excerpts), empirical identification of three pivotal quality dimensions (Dialogue Coherence, Character Consistency, Plot Reasonableness). Building upon these insights, we developed CML-Bench, an interpretable benchmark with eight quantitative metrics, and CML-Instruction, a structured prompting strategy designed to improve LLM-based script generation. Extensive experiments validate strong alignment with human preferences and demonstrate significant screenplay quality improvements, advancing creative AI for screenwriting.

ETHICS STATEMENT

Our CML-Bench framework is designed for academic research in screenplay evaluation and generation. The CML-Dataset derives from MovieSum under CC BY-NC 4.0 license, which we strictly follow. All LLM experiments use publicly available APIs in accordance with their terms of service. Human evaluation participants received fair compensation following regional ethical standards, with privacy rigorously protected. While our work enhances creative screenplay generation, we acknowledge potential environmental impact from computational resources and possible misuse of generative content. We commit to responsible dissemination through open-source releases (GitHub/HuggingFace) with clear academic-use guidelines. Dataset bias toward English-language classic films may limit cross-cultural applicability, which future work should address through multilingual expansion.

REPRODUCIBILITY STATEMENT

1. CML-Bench implementation details are described in Section 4; comprehensive metric formulations are provided in Equations (1)-(8).

2. CML-Dataset construction process is detailed in Section 3.1 and Appendix B. The complete dataset will be released on HuggingFace under CC BY-NC 4.0 license.

3. Source code for all metrics and evaluation pipelines will be available on GitHub under MIT license. Experimental hyperparameters and model configurations are detailed in Appendix F.

4. Human evaluation protocols and correlation analysis methods are described in Section 5 with statistical details in Appendix G.2.

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

Table 3: **Comparison of CML-Bench with other evaluation approaches, highlighting its specialized focus on cinematic narrative dimensions in screenplays.**

| Benchmark | Type | Focus | Domain Specificity |
|---|---|---|---|
| LongBench Bai et al. (2023b) | Metrics Analysis | Contextual Understanding | General Long Text |
| G-Eval Liu et al. (2023b) | LLM-as-a-judge | Holistic Scoring/Ranking | General NLG |
| EQ-Bench Paech (2023) | LLM-as-a-judge | Holistic Scoring/Ranking | General NLG |
| LoTBench Zhong et al. (2024) | LLM-as-a-judge | Multi-step Reasoning | General |
| **CML-Bench (Ours)** | Metrics Analysis & LLM-as-a-judge | **Cinematic Narrative Quality** | **Screenplay-Specific** |

# APPENDIX

## A LIMITATIONS AND FUTURE WORKS

**Limitations**  Our framework, while effective, has certain limitations. First, the CML-Dataset, while diverse with 600K tokens, is currently confined to 100 classic English-language films, which may not fully encompass the stylistic variety of contemporary or non-English screenplays. Additionally, CML-Bench metrics require a structured script format for analysis and might not be ideal for scripts written in a more free-form, natural language manner.

**Future Works**  Building on these limitations, future work will focus on two primary directions. First, we plan to extend the range and diversity of CML-Dataset and CML-Bench, incorporating a broader array of genres, contemporary scripts, and multilingual content to enhance robustness and applicability. Second, a crucial next step is to bridge the gap between script and screen by transferring script-level analysis to tangible movie-level visual outputs. This involves exploring methodologies to automatically generate visual representations (e.g., storyboards, character sketches, keyframes) from scripts and developing new metrics within CML-Bench to evaluate the potential visual impact and cinematic feasibility of generated screenplays.

## B STATISTICS ABOUT DATASET CONSTRUCTION

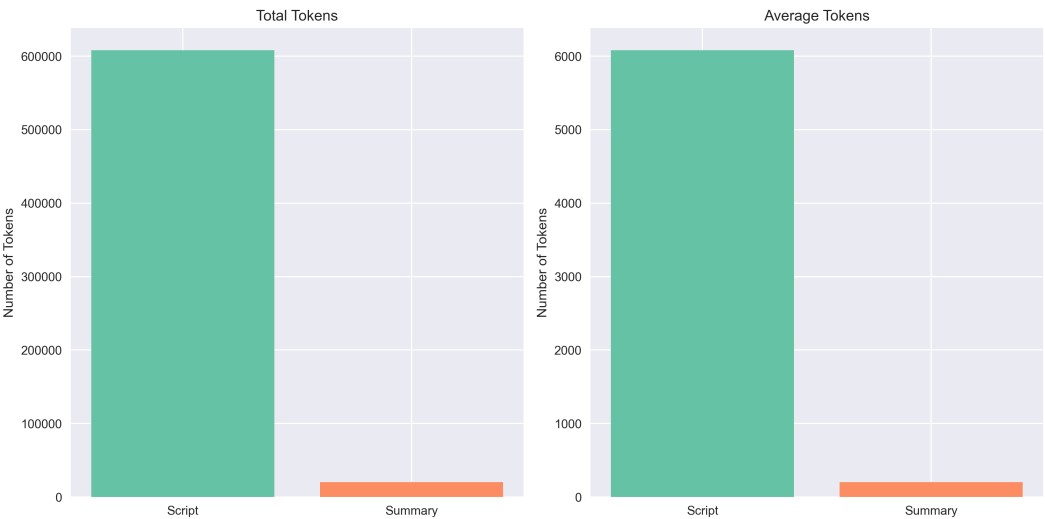

Figure 6: Overall token statistics for scripts and summaries in the CML-Dataset. The left panel shows total token counts, while the right panel shows average token counts.

**Overall Token Statistics.**  Figure 6 presents an overview of the token counts within the CML-Dataset. The left panel illustrates the total number of tokens, revealing a substantial volume for script content (approximately 600,000 tokens) compared to their corresponding summaries (around 20,000

tokens). The right panel shows that, on average, script segments contain roughly 6,000 tokens, while summaries are significantly more concise with an average of about 200 tokens.

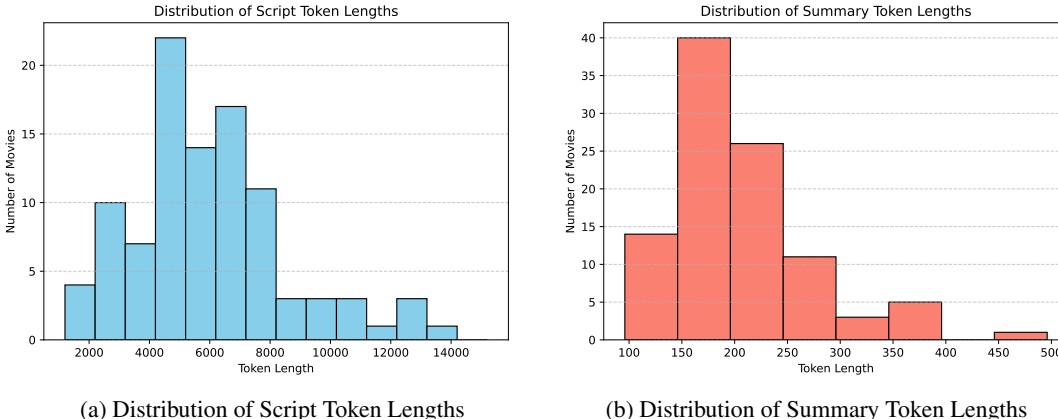

(a) Distribution of Script Token Lengths      (b) Distribution of Summary Token Lengths

Figure 7: **Token length distribution in CML-Dataset.** The dataset maintains text diversity while controlling length.

**Token Length Distributions.** The distributions of token lengths for script segments and their summaries are detailed in Figure 7. Panel (a) shows that script excerpts typically range from 1,500 to 14,000 tokens, with a concentration between 4,000 and 7,000 tokens, indicating diverse yet manageable complexity. Panel (b) demonstrates that summaries are consistently short, generally falling between 120 and 250 tokens, ensuring they are succinct inputs for generation tasks.

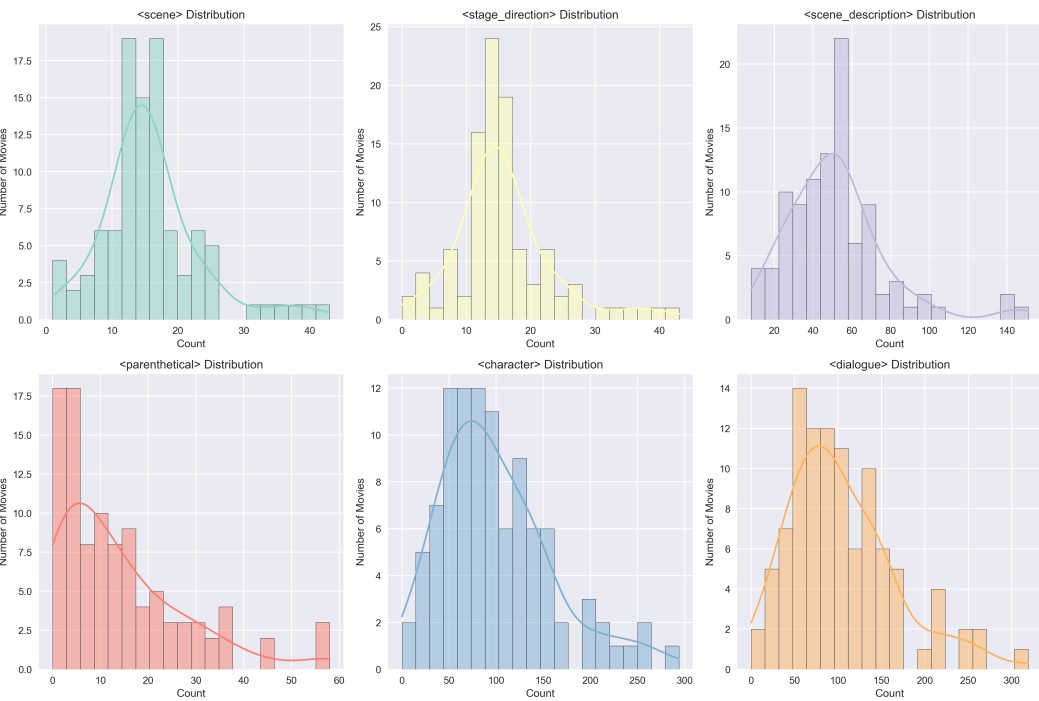

Figure 8: Distribution of common screenplay tags within the CML-Dataset script segments, including scene, stage direction, scene description, parenthetical, character, and dialogue tags.

**Screenplay Tag Distributions.** Figure 8 displays the frequency distributions for key screenplay tags across the CML-Dataset. The histograms illustrate common counts for elements such as `<scene>`, `<stage_direction>`, `<scene_description>`, `<parenthetical>`, `<character>`, and `<dialogue>` tags within the script segments. These distributions provide insights into the structural composition of the movie scripts included in our dataset.

**Quality Control Process.** Our systematic dataset construction approach employs a multi-stage quality control process to ensure high-quality screenplay content. We employed IMDb ratings 7.0 as our primary quality filter, ensuring selection of human-recognized high-quality scripts from critically acclaimed films. This approach leverages established human consensus on screenplay quality rather than subjective academic judgments. Our comprehensive approach included: (1) IMDb score-based filtering for human-validated quality, (2) professional screenplay format verification, (3) narrative completeness assessment, and (4) expert manual review for cinematic authenticity. This rigorous process ensures that our dataset maintains consistent quality standards while encompassing diverse cinematic styles and genres.

## C QUALITATIVE ANALYSIS EXAMPLES

### C.1 GOOD AND BAD CASE ANALYSIS

To demonstrate the effectiveness of our pre-set experiment in Section 3.2, we conducted LLM-as-a-judge analysis using Gemini 2.5 Pro to validate our intrinsic characteristics analysis. This analysis supports our framework's ability to distinguish screenplay quality through systematic evaluation of narrative logic and character authenticity.

**Low-Quality Case Analysis.** We examined "Dumb and Dumberer: When Harry Met Lloyd" (2003, IMDB: tt0329028), which represents a low-quality screenplay example. The following dialogue excerpt illustrates fundamental issues in plot reasonableness:

```
<character>LLOYD</character>
<dialogue>You could've saved that for the tooth fairy!</dialogue>
<character>HARRY</character>
<dialogue>That's stupid! I happen to know my mom is the tooth
fairy.</dialogue>
<character>LLOYD</character>
<dialogue>Your mom is the tooth fairy? That is so cool!</dialogue>
<character>HARRY</character>
<dialogue>Yeah, she must do all the flying around when
I'm asleep.</dialogue>
```

Gemini 2.5 Pro analysis identified this case as scoring 0.0 in plot reasonableness because adult characters treat childhood mythology (tooth fairy) as factual reality without narrative justification, demonstrating a fundamental breakdown in logical consistency.

**High-Quality Case Analysis.** In contrast, "Lake Placid" (1999, IMDB: tt0139414) exemplifies well-constructed screenplay dialogue:

```
<character>KEOUGH</character>
<dialogue>And what would he do come winter?</dialogue>
<character>HECTOR</character>
<dialogue>They can survive winter. As long as their nostrils
don't freeze,
they survive.</dialogue>
<character>HECTOR</character>
<dialogue>What I'm saying is if it's a crocodile that cut a man in half
he would have to be over twenty feet which would make him well over
a hundred years old, it would be unthinkable to destroy him.</dialogue>
```

This example scored 4.3/5 as characters demonstrate logical reasoning about environmental survival and biological constraints, maintaining professional discourse appropriate to their expertise. The dialogue exhibits coherent topic progression and character-consistent knowledge demonstration.

This comparative analysis confirms that our LLM-as-a-judge framework effectively captures the qualitative differences that distinguish high-quality screenplays from poorly constructed ones, validating our approach to automated screenplay evaluation.

# D    ADDITIONAL DETAILS OF BENCHMARKING

## D.1    PARSING RULES

The system uses a two-stage parsing strategy to extract elements from screenplays, prioritizing XML-like structures and falling back to plain text interpretation. This is primarily handled by the `parse_script_segment` function, with `parse_script_segment_base` as an alternative.

**Primary XML-like Parser.**    This parser targets scripts with explicit tags.

1. **Preprocessing:** Input is cleaned of common wrappers (e.g., " ```xml, `<script>`) and HTML entities are unescaped.
2. **Scene Segmentation:** Scripts are segmented into scenes using `<scene>`...`</scene>` tags. If no such tags are found, the entire script is treated as a single scene/action block.
3. **Element Extraction:** Within scenes, content is extracted from `<character>`, `<dialogue>`, `<parenthetical>`, `<action>`, `<scene_description>`, and `<stage_direction>` tags. Nested tags within these are removed.
4. **Data Collection:** Dialogues are ordered and grouped by character, with parentheticals linked to their respective dialogue. Actions and scene descriptions (including initial stage directions) are collected; these descriptions also inform scene-level coherence metrics (e.g., PR1).

**Fallback Plain Text Parser.**    This parser is used for less structured inputs.

1. **Scene Segmentation:** Text is split into scenes based on multiple empty lines or keywords like "INT."/"EXT."
2. **Dialogue and Action Extraction:** Dialogues are identified by "CHARACTER NAME: Text" patterns. Remaining text within a scene segment is considered an action.

This dual approach ensures that structural elements crucial for metric calculation are extracted from varied script formats, directly impacting the data available for quality assessment.

## D.2    EXPLAINING ZERO SCORES FOR SOME LLMS

Several Large Language Models (LLMs), particularly when used with basic prompting ("base" versions), may generate outputs that result in zero or near-zero scores for multiple CML-Bench metrics. This is often not an indictment of the metric's sensitivity but rather a direct consequence of the LLM's failure to produce a structurally coherent and parsable screenplay. The parsing rules, detailed in Section D.1, are designed to extract specific screenplay elements, and if these elements are absent or incorrectly formatted, the data required for metric calculation cannot be obtained.

**Impact of Poor Structure on Parsing.**    As defined in Sec 3, movie scripts are highly structured data types. Consider an output from a model like "o3-mini" under base prompting conditions. A typical failure mode involves the generation of text that lacks clear delimitation of scenes, character cues, or dialogue. The primary parser would find no `<scene>`, `<character>`, or `<dialogue>` tags. The fallback parser might struggle if character cues do not strictly adhere to the "CHARACTER: Text" format or if scene boundaries are ambiguous. Consequently, the parsing process might yield empty or sparsely populated lists for "dialogues ordered", "dialogues by character", "scenes", and "actions".

**Consequences for Metric Scores.**    The absence of correctly parsed elements directly leads to low or zero scores for several metrics:

- **Dialogue Coherence (DC):**
    - For DC1 (Adjacent Turn Topic Similarity, Equation 1 in Section 4), if fewer than two dialogue turns are extracted ("dialogues ordered" is too short), the metric defaults to 0.
    - For DC2 (Dialogue Topic Concentration, Equation 2), a lack of dialogue means no keywords can be extracted, resulting in a score of 0.
    - For DC3 (Question-Answer Pair Relevance, Equation 3), no dialogues mean no Question-Answer pairs can be identified, leading to a 0 score.
- **Character Consistency (CC):**

- For CC1 (Emotional Stability, Equation 4) and CC2 (Linguistic Style Consistency, Equation 5), if a character has fewer than two or three dialogue turns parsed respectively ("dialogues by character"), these metrics cannot be computed for that character, lowering the overall average. If no characters have sufficient dialogue, the scores become 0.
    - For CC3 (Action-Intention Alignment, Equation 6), if no dialogues (for intentions) or no actions are parsed, the score is 0.
- **Plot Reasonableness (PR):**
    - For PR1 (Event Sequence Semantic Coherence, Equation 7), if fewer than two scenes are parsed, the score is 0.
    - For PR2 (Causality Density, Equation 8), if no scenes or actions are extracted to form a basis for event identification $K = 0$, leading to a 0 score. However, we found that PR2 did not appear to be 0, while Table 1 and Figure 4 also show that PR2 is not a good indicator of discrimination.

**Contrast with Instruction-Tuned Output.** In contrast, an instruction-tuned model like "o3-mini-instruction", when guided by CML-Instruction (Section E), typically produces output that adheres to the expected XML-like structure. As a result, all CML-Bench metrics can be computed, leading to more meaningful (non-zero) scores that reflect the actual quality of the generated content rather than a parsing failure. The significant performance gap often observed between "base" and "instruction" versions of models in Table 1 can thus be substantially attributed to the improvement in structural adherence guided by CML-Instruction, which enables successful parsing and subsequent metric evaluation.

# E CML-INSTRUCTION

## E.1 PROMPT ENGINEERING FOR *Base* SUMMARIES

Our goal in the first stage is to obtain a concise yet information-dense *summary* $a_i$ for every 15–20-scene script segment $c_i$ (cf. Section 3.1). We found that a single "structured-bullet" prompt works robustly across genres and lengths.

## E.2 PROMPT ENGINEERING FOR INSTRUCTION SCRIPT GENERATION

CML-Instruction is a complex prompt designed to guide an LLM to expand a given movie summary into a well-structured screenplay segment. In our implementation, the final instruction prompt is constructed by concatenating four key string components: *"prompt start llm'", "prompt instructions content", "prompt example", and "prompt end llm"*. These components are detailed below. (Note: The overall LLM interaction typically begins with a system message like "You are an award-winning screenwriter," which sets the persona before this concatenated user prompt is provided.)

COMPONENT 1: STARTING DIRECTIVE (*prompt start llm*)

This component sets the expert role for the AI and introduces the primary task, including placeholders for the movie title and summary that are filled at runtime.

```
You are an expert AI scriptwriter. Your task is to generate a detailed
    and professional movie script segment based on the provided Movie
    Title and Movie Summary. The script should be formatted in an XML-
    like structure, mirroring professional screenplay standards.

**Input:**
Movie Title: {movie_name}
Movie Summary: {summary}
```

Listing 1: Content of "prompt start llm".

COMPONENT 2: DETAILED CONTENT INSTRUCTIONS (*prompt instructions content*)

This part provides specific rules for script generation, covering overall structure, scene elements, and content/style guidelines. These instructions are designed to align the generated script with the quality dimensions evaluated by CML-Bench.

```
**Instructions for Script Generation:**

1. **Overall Structure:**
   * The entire script segment must be enclosed within `<script> ... </
      script>` tags.
   * The script should be divided into multiple `<scene> ... </scene>`
      blocks.

2. **Scene Elements (within each `<scene>`):**
   * **Stage Direction (Scene Heading):** Start each scene with a `<
      stage_direction>...</stage_direction>` tag. This should specify
      the location (INT. or EXT.), the specific place, and the time (DAY,
       NIGHT, CONTINUOUS, etc.). For example: `<stage_direction>INT.
      POLICE STATION - DAY</stage_direction>`.
   * **Scene Description:** Use `<scene_description>...</
      scene_description>` tags for detailed narrative descriptions. This
       includes:
      * Setting details (visuals, atmosphere).
      * Character actions, movements, and significant non-verbal
         expressions.
      * Key sounds or visual cues.
      * The flow of events within the scene.
   * **Character Dialogue:**
      * Introduce a speaking character with `<character>CHARACTER NAME</
         character>` tag (character names are typically in ALL CAPS).
      * Follow with their speech in a `<dialogue>...</dialogue>` tag.
      * If a character has brief acting notes or delivery instructions
         directly related to their dialogue, use a `<parenthetical>(note)
         </parenthetical>` tag immediately before or interspersed within
          their `<dialogue>` as appropriate. For example, `<
         parenthetical>(whispering)</parenthetical>` or `<parenthetical
         >(V.O.)</parenthetical>`.
   * **Actions/Further Descriptions within Scenes:** Additional `<
      scene_description>` tags can be used within a scene to describe
      actions that occur between dialogues or to elaborate further on
      the ongoing scene. Ensure these descriptions are vivid and
      contribute to the scene's progression.

3. **Content and Style:**
   * The generated script segment should logically follow from the
      provided Movie Summary, developing key events and character
      interactions implied by it.
   * Maintain consistency in character voice, behavior, and motivations
      throughout the segment.
   * Ensure dialogue is natural, engaging, and serves both plot
      advancement and character development.
   * Scene descriptions should be vivid, concise, and provide enough
      detail for visualization, focusing on what can be seen and heard.
   * The script should be coherent, with smooth and logical transitions
      between descriptions, actions, and dialogues.
   * Focus on creating a script segment that feels like a continuous and
       integral part of a larger, professional screenplay.
   * The tone and style should match that of a production-ready script.
```

Listing 2: Content of "prompt instructions content".

COMPONENT 3: ILLUSTRATIVE EXAMPLE (*prompt example*)

An example snippet is provided to further clarify the expected XML-like format and the interplay of different tags.

```
**Example Snippet of Expected Format:**
```xml
```

```
<script>
  <scene>
    <stage_direction>INT. COFFEE SHOP - DAY</stage_direction>
    <scene_description>The coffee shop is bustling. ANNA (30s), dressed in
        a sharp business suit, sips her latte, looking impatient. MARK
        (30s), disheveled and out of breath, rushes in.</scene_description>

    <character>MARK</character>
    <dialogue>Sorry I'm late! The traffic was insane.</dialogue>
    <character>ANNA</character>
    <parenthetical>(glancing at her watch)</parenthetical>
    <dialogue>Insane or you overslept?</dialogue>
    <scene_description>Mark pulls out a chair and slumps into it, running
        a hand through his messy hair. He looks exhausted.</
        scene_description>
    <character>MARK</character>
    <dialogue>Okay, a bit of both. But mostly insane traffic.</dialogue>
  </scene>
  <scene>
    <stage_direction>EXT. PARK - LATER</stage_direction>
    <scene_description>Sunlight dapples through the trees. Anna and Mark
        walk along a paved path, a little more relaxed now.</
        scene_description>
    <character>ANNA</character>
    <dialogue>So, about the Henderson account... We need a new strategy.</
        dialogue>
  </scene>
</script>
```
```

Listing 3: Content of "prompt example".

COMPONENT 4: FINAL COMMAND (*prompt end llm*)

This concluding part explicitly instructs the LLM to generate the script based on the provided inputs and guidelines, reinforcing the desired output format.

```
Please generate the script segment based on the Movie Title and Summary
    provided above, adhering strictly to this XML-like format and content
     guidelines. Ensure the output is a single block of text starting
    with `<script>` and ending with `</script>`.
```

Listing 4: Content of "prompt end llm".

# F    EXPERIMENT SETTINGS

All experiments and metric computations were conducted on a single server node equipped with eight NVIDIA H100 GPUs. The calculation of our proposed CML-Bench metrics utilized two primary large language models: Qwen QwQ 32B Yang et al. (2024) and Gemma-2-2B Team et al. (2024b). Specifically, the Qwen QwQ 32B model, when employed for certain complex metric calculations (e.g., those requiring deep contextual understanding or causal reasoning), processed input sequences of up to 55K tokens. For these tasks, the Qwen model typically required approximately 36GB of VRAM in two H100 GPUs. Evaluating a full set of scripts generated by the seven baseline LLMs (as detailed in Table 1 in the main paper) using two H100 GPUs took approximately 2 hours to complete. The Gemma-2-2B model was utilized for lighter-weight tasks such as embedding generation and keyword extraction, offering efficient processing.

## G   ADDITIONAL DETAILS OF USER STUDY

### G.1   INSTRUCTIONS FOR HUMAN EVALUATION RATING

Human experts were tasked with evaluating script excerpts generated by various models and human-written ground truth. To guide this process, evaluators were presented with a standardized interface, examples of which are illustrated in Figure 9. This figure shows how movie script segments were paired with rating scales for three key dimensions: Dialogue Coherence (DC), Character Consistency (CC), and Plot Reasonableness (PR).

Participants, comprising ten individuals with experience in narrative assessment, rated excerpts from five randomly sampled movies. For each excerpt, they assigned scores from 0 (very poor) to 5 (excellent) for each of the three dimensions based on the following criteria:

- **Dialogue Coherence (DC):** Evaluates the logical flow, naturalness, and topical consistency of conversations. A high score indicates that dialogue is easy to follow, relevant to the context, and maintains a clear purpose.
- **Character Consistency (CC):** Assesses the stability of characters' linguistic styles, emotional expressions, and motivations throughout the excerpt. A high score means characters behave and speak in a way that is consistent with their established persona.
- **Plot Reasonableness (PR):** Judges the plausibility and logical progression of events and character actions within the narrative. A high score reflects a storyline that is coherent and whose developments are well-motivated.

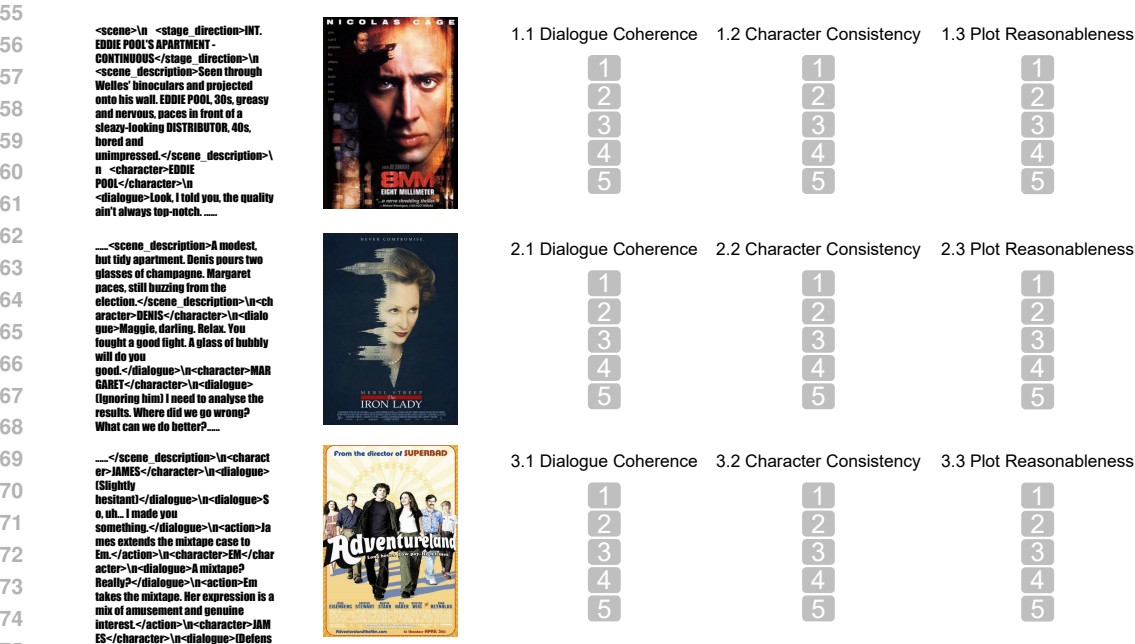

Figure 9: **Demonstration of the interface elements shown to human evaluators**. For each movie (represented by its poster), experts rated script excerpts on Dialogue Coherence, Character Consistency, and Plot Reasonableness using a 0-5 scale for each dimension.

### G.2   DETAILED CORRELATION ANALYSIS WITH HUMAN JUDGMENTS

To quantify the agreement between CML-Bench and human evaluations, we performed a correlation analysis using the Spearman rank correlation coefficient ($\rho$). This non-parametric measure assesses the strength and direction of a monotonic relationship between two ranked variables. It is suitable when the data may not be normally distributed or when the relationship is not strictly linear but consistently increasing or decreasing.

The Spearman correlation coefficient is calculated by first converting the scores for each variable (e.g., average CML-Bench scores and average human ratings for each script source listed in Table 2)

into ranks. Then, the Pearson correlation coefficient is computed on these ranks. Alternatively, if there are no tied ranks, it can be calculated using the formula:

$$\rho = 1 - \frac{6 \sum d_i^2}{n(n^2 - 1)}$$

where $d_i$ is the difference between the two ranks for each observation (script source), and $n$ is the number of observations (15 script sources in our case, including Human Ground Truth, 7 base LLMs, and 7 instruction-tuned LLMs).

First, for each of the 15 evaluated script sources, an average CML-Bench score was calculated by taking the mean of its scores for Dialogue Coherence (DC), Character Consistency (CC), and Plot Reasonableness (PR), as detailed in Table 2. Second, a corresponding average human rating was calculated for each script source by averaging the human-assigned scores for DC, CC, and PR, also from Table 2.

The analysis yielded an overall Spearman correlation coefficient of $\rho = 0.80$ with a p-value of 0.0000. A p-value this small indicates that the observed correlation is statistically significant and unlikely to have occurred by chance. This strong positive correlation ($\rho$ close to 1) suggests that as the CML-Bench scores for screenplay quality increase, human ratings also tend to increase, confirming a high degree of monotonic agreement between our automated benchmark and averaged human perceptions of overall screenplay quality. This reinforces the benchmark's utility in reflecting human-like evaluation judgments for script assessment.

## H  THE USAGE OF LARGE LANGUAGE MODELS

Large Language Models play a fundamental role in multiple aspects of our CML-Bench framework, extending beyond traditional text processing to core methodological contributions:

**Dataset Creation and Curation.**  LLMs are integral to our dataset construction process. We employ Qwen large language models to automatically identify and extract coherent 15-20 scene segments from lengthy movie scripts, ensuring each excerpt forms a complete narrative unit. This automated segmentation process leverages LLMs' understanding of narrative structure to maintain story coherence while creating manageable evaluation units. Additionally, LLMs assist in format standardization and quality verification of screenplay content during the curation process.

**Novel LLM-as-Info Benchmarking Framework.**  Our core methodological innovation involves using LLMs as information extractors rather than direct judges. Specifically, we employ Gemma-2-2b-it for lightweight feature extraction (keywords, creative language features, character traits) and Qwen models for complex semantic analysis and embedding generation. This dual-LLM architecture enables sophisticated feature extraction while maintaining computational efficiency. The extracted information is then processed through our quantitative metrics that combine similarity and dissimilarity calculations to assess both coherence (similarity-based) and creativity (dissimilarity-based) aspects of screenplays.

**Metric Design and Validation.**  LLMs contribute to the design and validation of our nine evaluation metrics across Dialogue Coherence (DC), Character Consistency (CC), and Plot Reasonableness (PR) dimensions. For instance, in our creativity metrics (DC3: Linguistic Creativity, CC2: Character Originality, PR3: Narrative Innovation), LLMs analyze creative elements and generate embeddings that are subsequently processed through cosine similarity calculations with 1-x transformations to measure novelty and originality.

**Experimental Evaluation and Analysis.**  LLMs are employed in our comparative analysis framework, including the intrinsic characteristics analysis that validates our three core dimensions. Gemini 2.5 Pro is specifically used for qualitative analysis of good versus bad screenplay examples, providing detailed reasoning about plot logic, character consistency, and dialogue coherence that supports our quantitative findings.

**Traditional Research Support.** Beyond these methodological contributions, LLMs assist in conventional research activities including grammar checking, format optimization of figures and tables, and language polishing for clarity and academic style. However, all content generated by LLMs undergoes thorough human review and validation.

**Responsibility and Oversight.** All authors maintain full responsibility for LLM-generated content. Every output is carefully reviewed, validated, and integrated into our research framework with appropriate human oversight. The LLM contributions are designed to enhance rather than replace human expertise in screenplay analysis and evaluation methodology development.

