# OpenReview forum: "CML-Bench: A Framework for Evaluating and Enhancing LLM-Powered Movie Scripts Generation"
_ICLR.cc/2026/Conference — Submitted to ICLR 2026_

### Official Review · Reviewer_DnJr · 2025-10-21

**Soundness:** 2
**Presentation:** 1
**Contribution:** 1
**Rating:** 2
**Confidence:** 5

**Summary:**

This paper aims to address the common failure of LLMs to generate movie scripts with sufficient emotional depth and narrative coherence. To do this, the authors first introduce the CML-Dataset, a collection of 100 script segments from highly-rated films paired with LLM-generated summaries. Based on an analysis of these scripts, they identify three core dimensions for quality assessment: DC, CC and PR. They then build upon these dimensions to create CML-Bench, a quantitative evaluation benchmark, and propose CML-Instruction, a structured prompting strategy using XML-like tags to improve script generation.

**Strengths:**

- This paper is easy to follow.
- The paper tackles an important problem. Evaluating the quality of long-form creative text like screenplays is far more complex than standard NLP tasks.

**Weaknesses:**

**1. Internal Inconsistencies and Lack of Polish**

The manuscript contains contradictory statements and appears unfinished. Throughout the paper, it sometimes states that the benchmark has 8 metrics, and at other times it says 9. This caused significant confusion for the reader. For example, L88 mentions that CML-bench consists of 8 interpretable metrics, but L251 says that CML-bench has 9 interpretable metrics.

Furthermore, the manuscript contains numerous unresolved references, which appear as double question marks. For example, L95, L106 and L238. This suggests a lack of careful proofreading before submission and significantly undermines the paper's professionalism.

Table 1 presents numerical results with inconsistent precision, mixing values with two and three decimal places without clear justification. This suggests a lack of attention to detail in the presentation of results.

**2. CML-Instruction is insufficient to be considered a key contribution**

The paper presents CML-Instruction as a key contribution. However, its core mechanism is a form of in-context learning, providing detailed instructions and examples in the prompt, which is a well-established capability of basically all modern LLMs. Claiming this widely supported prompting technique as a novel contribution overstates its originality.

**3. Experimental Evaluation Confounds Formatting with Quality**

Another main flaw of the paper lies in its experimental design. The results in Table 1 show that base LLMs score very poorly, while instruction models score close to human level. However, the appendix candidly admits that the primary reason for the low scores is the failure of base models to generate the specific XML-like format expected by the CML-Bench parser. Therefore, the experiment does not robustly demonstrate that CML-Instruction improves the creative quality of the script, but rather that it improves the model's adherence to a specific parser format.

**4. Confusing Metrics**

Some of the metric definitions appear very confusing. For example, the descriptions for PR1 (Event Sequence Semantic Coherence) and PR2 (Event Coherence) make the me feel they are nearly identical. And there is no formulation for PR2 to help readers better understanding it. Furthermore, the authors admit in the appendix L982 that "PR2 is not a good indicator of discrimination".
This admission is confusing and directly contradicts the PR2's inclusion, which severely undermines the credibility of CML-Bench as a carefully designed benchmark.

**5. Not Enough Samples**

The CML-Dataset contains only 100 samples. This is a very small number for a benchmark intended to evaluate a complex, high-variance task like creative script generation.

**Questions:**

Please see weaknesses.

---

### Official Review · Reviewer_gRNh · 2025-10-28

**Soundness:** 2
**Presentation:** 2
**Contribution:** 3
**Rating:** 2
**Confidence:** 4

**Summary:**

This paper contributes CML-Dataset, CML-Bench, and CML-Instruction. The CML-Dataset consists of script-summary pairs where the script is a 15-20 scene segment of a popular movie script and the summary is generated by an LLM describing this script. Using this dataset, they then design 9 metrics which can be used to judge dialogue coherence, character consistency, and plot reasonableness of scripts. These metrics in combination with their dataset form CML-Bench, a benchmark on which to judge LLM abilities to generate high quality movie scripts. Finally, they propose CML-Instruction, a method of prompting LLMs which improves the quality of their generated movie scripts.

**Strengths:**

Originality:
- the proposed metrics seem to distinguish between high quality and low quality scripts well and provide a nice combination of structured and LLM-based components

Quality:
- a human study is included to validate results

Clarity:
- the visual depictions of results help to clearly communicate performance details
- contributions are clear

Significance:
- the proposed metrics seem like they would be useful to help evaluate future movie script generation methods

**Weaknesses:**

1. The presentation of the paper lacks polish. For example, the intro has a broken figure reference, there are several missing periods throughout, the wrong \cite command is frequently used (citations should be in parentheticals in most places), figure captions lack important details (are the numbers in figure 2 counts or percents? what is the x axis in figure 3 and are the values averages or singular examples?), and the font in many plots is too small to be legible (e.g., figure 5).

2. There are not enough details of the human study to determine its validity. Most importantly, who are the human annotators and how were they recruited? If applicable, was IRB approval obtained? What is the level of agreement between annotators?

3. There is not enough detail on the metric implementation to fully understand them. Most importantly, what is the creativity analysis noted in line 281? Also, how are emotions classified? Finally, there are many points where an LLM is used to extract some detail (e.g., keywords, distinctive speech features, etc.) and we need more details on how this is done to understand the output of these extractions.

4. I am skeptical of the usefulness of CML-Instruction and CML-Bench. Figure 4 shows that CML-Instruction brings LLMs mostly up to ground-truth performance across almost all metrics. If this is the case, is performance on this benchmark already saturated? Also, why does CML-Instruction improve results so much across all metrics? Without more qualitative discussion of these results, I'm skeptical that the methodology is not somehow gaming the metrics. In general, the results section needs more discussion of the results as it is currently just one short paragraph of text.

**Questions:**

Suggestion: The intro includes a lot of detail on the dataset collection and methods that could be left for the later section. The intro would be stronger if it focused more on contextualizing the importance of the work and key takeaways.

---

### Official Review · Reviewer_yFrW · 2025-10-31

**Soundness:** 3
**Presentation:** 3
**Contribution:** 3
**Rating:** 4
**Confidence:** 3

**Summary:**

This paper presents CML-Bench, a comprehensive framework for evaluating and enhancing movie script generation by large language models (LLMs). Although LLMs are proficient at producing structurally coherent text, they often fall short in capturing the narrative depth and emotional nuance essential to cinematic storytelling. To bridge this gap, the authors introduce a pipeline comprising three key components: CML-Dataset, CML-Bench, and CML-Instruction. Experimental results show that while LLMs struggle across all screenplay quality dimensions, applying CML-Instruction leads to significant improvements, yielding scripts with coherence and logic comparable to human-written ones. Human evaluations further demonstrate strong alignment with CML-Bench scores, validating the benchmark’s robustness and reliability.

**Strengths:**

1. This work introduces a well-structured, domain-specific benchmark (CML-Bench) with interpretable metrics and a curated dataset (CML-Dataset), filling a clear gap in evaluating LLMs for creative, long-form movie script writing.

2. The CML-Instruction prompting strategy is practical and demonstrates strong, quantitative improvements on various LLMs, as shown in Figure 4 and Table 1.

3. Main results on CML-Bench are comprehensive, covering a wide range of mainstream LLMs (e.g., Qwen, LLaMA, Gemini, DeepSeek), which ensures the conclusions are broadly representative and robust across model architectures.

4. The paper is clearly written and well-organized.

**Weaknesses:**

1. The CML-Dataset remains relatively small (100 films) and limited to English-language, high-rated classic films, which may restrict its generalizability.

2. The evaluation still partially depends on LLM-based feature extraction, which may introduce bias.

3. The comparison and discussion with existing benchmarks (e.g., MovieBench, Movie101) is missing, making it difficult to assess CML-Bench's contribution to the community.

4. The caption in Figure 5 is incomplete, and the illustration is too blurry to read.

5. L107: Missing figure reference number.

**Questions:**

The performance improvements achieved with CML-Instruction are substantial across all evaluated models, in some cases approaching human-level results, but the performance on character originality remains consistently low across models. Could the authors discuss what factors might contribute to this discrepancy?

**Details Of Ethics Concerns:**

The paper may raise fairness and bias concerns since the CML-Dataset is composed exclusively of English-language, high-rated classic films, which could reinforce cultural and stylistic biases in both evaluation and generation.

---

### Official Review · Reviewer_AWtP · 2025-11-02

**Soundness:** 3
**Presentation:** 2
**Contribution:** 2
**Rating:** 4
**Confidence:** 4

**Summary:**

The paper introduces CML-Bench (Cinematic Markup Language Benchmark) — a new benchmark framework for evaluating large language models (LLMs) on screenplay or movie script generation. Unlike traditional text generation benchmarks, CML-Bench focuses on cinematic storytelling quality, aiming to measure whether LLMs can produce coherent, character-consistent, and narratively reasonable scripts. The benchmark design is logically organized into three principal evaluation dimensions—Dialogue Coherence (DC), Character Consistency (CC), and Plot Reasonableness (PR)—each further decomposed into mathematically defined sub-metrics. As a benchmark work, this paper design comprehensive evaluation metrics and comparative baselines to give a solid results. However, there is room to improve on presentation. Here are some detailed comments:
1. Lack of clear definition of "Cinematic Markup Language”
The concept of Cinematic Markup Language—which appears to be central to this paper—is not clearly defined or contextualized in the introduction or related work sections. Since this is not a widely recognized or established task in the broader multimodal learning community, readers unfamiliar with the term will find it difficult to understand the problem formulation and motivation. I strongly recommend that the authors provide a precise and intuitive definition of CML early in the paper (e.g., in the introduction or Section 2), clarify its distinction from related multimodal representation or video annotation tasks, and perhaps add one or two concrete examples for better accessibility.
2. Formatting and figure quality issues.
The manuscript contains several writing and formatting errors that significantly affect readability. For example: There are multiple broken or missing figure references on page 2 and page 5 (e.g., "Figure ??"), suggesting incorrect LaTeX linking. Several figures (e.g., Figures 1, 2, and 4) have font sizes that are too small to read clearly, especially in axis labels and legends.
These presentation issues make it difficult for readers to interpret the experimental setup and key results. The authors should carefully proofread the paper, fix the figure reference errors, and increase the resolution and font size of all plots and diagrams in the future version.

**Strengths:**

1.	Addresses an underexplored but valuable task. The paper tackles an emerging and underrepresented domain — screenplay generation and cinematic storytelling, which indeed deserves attention within the multimodal and creative AI community.
2.	Conceptual originality in metric design. The decomposition of screenplay quality into nine measurable components is creative and well-aligned with the narrative theory of scriptwriting, offering a potentially reusable evaluation framework for creative generation tasks.

**Weaknesses:**

1.	Questionable validity of benchmark effectiveness. A major concern is that several LLM baselines outperform human-written scripts (the supposed “ground truth”) on many of the proposed metrics. The authors may could provide a qualitative or human-evaluation-based sanity check to justify this outcome.
2.	Insufficient evidence of real-world applicability. It remains unclear whether the proposed benchmark will generalize beyond this dataset or meaningfully correlate with human judgments.
3.	Writing quality. I recommend authors polish the paper for better understanding.

**Questions:**

The questions are similar with what I mentioned in the weakness part. There are mainly two questions:
1.	Is the usage of MovieSum’s training set reasonable? These movies are classic and famous movies, there might be some data leak for those LLMs.
2.	The effectiveness of benchmark. The score of many LLMs are better than GT.

---

### Meta-Review · Area_Chair_c2Jf · 2026-01-21

**Summary:**

Methodological Concerns:
- Benchmark Validity: Several LLMs outperform human-written "ground truth" scripts on proposed metrics, raising questions about whether the benchmark truly measures quality and whether this benchmark is already saturated or still useful. (AWtP, gRNh)
- Evaluation Design Flaw: Low base model scores primarily reflect failure to generate expected XML format rather than poor creative quality (DnJr)
- Metric Confusion: PR1 and PR2 appear nearly identical; authors admit PR2 is "not a good indicator of discrimination" (DnJr)

Dataset Limitations:
- Only 100 samples—too small for high-variance creative tasks (yFrW, DnJr)
- English-only, limited to classic high-rated films—generalizability concerns (yFrW)
- Potential data leakage since MovieSum scripts may be in LLM training data (AWtP)

Contribution Concerns:
- CML-Instruction is essentially in-context learning, a well-established technique—limited novelty (DnJr)
- Missing comparison with existing benchmarks like MovieBench, Movie101 (yFrW)
- Important human study details (recruitment, IRB, inter-annotator agreement) (gRNh)

**Reviewer Concerns:**

The author does not participate in the rebuttal phase.

**Reviewer Scores:**

Probably not changed based on 2 reasons: 1) there is no response from the author; 2) most of the proposed concerns are structural issues difficult to address in rebuttal.

---

### Decision · Program_Chairs · 2026-01-26

Reject